# Influence of Population Size, the Human Development Index and the Gross Domestic Product on Mortality by COVID-19 in the Southeast Region of Brazil

**DOI:** 10.3390/ijerph192114459

**Published:** 2022-11-04

**Authors:** Mônica Feresini Groppo, Francisco Carlos Groppo, Sidney Raimundo Figueroba, Antonio Carlos Pereira

**Affiliations:** 1Community Dentistry Department, Piracicaba Dental School, University of Campinas—UNICAMP, Av. Limeira, 901, Bairro Areião, Piracicaba 13414-903, SP, Brazil; 2Department of Biosciences, Piracicaba Dental School, University of Campinas—UNICAMP, Av. Limeira, 901, Bairro Areião, Piracicaba 13414-903, SP, Brazil

**Keywords:** COVID-19, epidemiology, Brazil, HDI, GDP

## Abstract

We evaluated the influence of population size (POP), HDI (Human Development Index) and GDP (gross domestic product) on the COVID-19 pandemic in the Southeast region of Brazil, between February 2020 and May 2021. Methods: Cases, deaths, incidence coefficient, mortality rate and lethality rate were compared among states. The cities were divided into strata according to POP, GDP, and HDI. Data were compared by Welch’s ANOVA, nonlinear polynomial regression, and Spearman’s correlation test (rS). Results: The highest incidence coefficient (*p* < 0.0001) and mortality rate (*p* < 0.05) were observed in the states of Espírito Santo and Rio de Janeiro, respectively. Until the 45th week, the higher the POP, the higher the mortality rate (*p* < 0.01), with no differences in the remaining period (*p* > 0.05). There was a strong positive correlation between POP size and the number of cases (rS = 0.92, *p* < 0.0001) and deaths (rS = 0.88, *p* < 0.0001). The incidence coefficient and mortality rate were lower (*p* < 0.0001) for low GDP cities. Both coefficients were higher in high- and very high HDI cities (*p* < 0.0001). The lethality rate was higher in the state of Rio de Janeiro (*p* < 0.0001), in large cities (*p* < 0.0001), in cities with medium GDP (*p* < 0.0001), and in those with high HDI (*p* < 0.05). Conclusions: Both incidence and mortality were affected by time, with minimal influence of POP, GDP and HDI.

## 1. Introduction

On 30 January 2020, the World Health Organization (WHO) declared an Emergency in Public Health of International Interest and officially announced it as a “world pandemic” about two months later [1]. Despite efforts in several countries to control the pandemic, global growth in the number of cases and deaths continued to increase [2].

The coronavirus of severe acute respiratory syndrome (SARS) type 2 or SARS-CoV-2 has the potential to contaminate humans, mammals and birds, causes liver, enteric, neurological, and respiratory diseases [1], and is transmitted by droplets or respiratory aspersions. Symptomatology starts with fever, dry cough, and dyspnea. It may progress to pneumonia, SARS, and death, but the asymptomatic form is more common. Asymptomatic cases can cause high underestimation of cases in official statistics, since asymptomatic people do not seek medical attention [3].

In Brazil, the first case of the disease was reported on 26 February 2020, in a city in the state of São Paulo [4] and rapidly spread, with distinct regional differences. Some factors, such as the regional heterogeneity, different temporal dissemination, and disparities in the clinical management practices, contributed to the differences in pandemic progression verified among the regions of Brazil [5]. In August 2020, Brazil began to report a faster rate of cases and deaths than the global average [1].

At the beginning of the pandemic, the accumulated number of cases and deaths was higher in the North, Northeast, and Southeast regions, probably due to the low Human Development Indexes (HDI), the highest proportions of poverty, and low level of education in the North and Northeast regions. In addition to regional inequalities, Brazil faced additional difficulties to control the pandemic, such as weakening of preventive measures, insufficient health care infrastructure, high dependence on federal economic assistance, high proportion of elderly with comorbidities and lack of social isolation, leading to the country becoming the epicenter of the disease in 2021 [6].

The Unified Health System (SUS) is responsible for primary health care in Brazil. This system is highly decentralized, and it is present in the places closest to those where the citizens live. It is the initial and preferential form of contact in public health care. Moreover, it is the gateway to and the communication center for the entire national health care network [7]. 

Despite the general efficiency of this health system, there are some structural problems in relation to governance, organization, and financing that still need to be solved [5]. In fact, inequalities increased during the pandemic, generating the risk of different outcomes in individuals with the same level of morbidity [8].

The Brazilian Southeast region has the largest number of health plan beneficiaries, but it has experienced rapid and exaggerated evolution of cases and deaths [1] on more than one occasion since March 2020.

Brazil has approximately 450,000 hospital beds, with almost 315,000 (70%) belonging to the SUS. The Southeast region alone contains about 41% of all the hospital beds in Brazil, followed by the Northeast (27%), Southern region (16%), Midwest (8.5%) and North region (7.3%). Notably, the Southeast region contains the largest population (around 80 million people or 42% of all Brazilians). In addition, it is undoubtedly the richest region in the country and it presents the highest urbanization indexes, and the largest industrial zone, being responsible for more than half of Brazilian GDP [1]. Despite these characteristics, the Southeast region has many disparities regarding the distribution of the population, economics, and human development, meaning that it is representative of the Brazilian regional differences.

To date, there are no studies that show how the pandemic was affected in the states of the Southeast region, or on the influence of population size, economy, or human development in terms of evolution of the pandemic. Thus, the aim of this study was to evaluate the impact of the population size, the economy (measured by municipal GDP per capita) and Municipal Human Development Index (measured by HDI-M) of the cities of the Brazilian Southeast region, on the cases of COVID-19 and mortality due to the COVID-19 pandemic, between 25 February 2020 and 29 May 2021.

## 2. Materials and Methods

### 2.1. Ethical Aspects

All data were obtained from freely accessible online databases. Consolidated data on cases/deaths, and population estimates were obtained from the health information systems in the “Coronavirus Panel” of the Brazilian Ministry of Health (https://covid.saude.gov.br/, accessed on 30 May 2021).

### 2.2. Variables

Only the four states (*Espírito Santo*, *Minas Gerais*, *Rio de Janeiro*, and *São Paulo*) of the Brazilian Southeast region were included in the study.

The population size (POP) of officially recognized cities (cities that have a city hall) was divided into a long list of classifications by the Brazilian Institute of Geography and Statistics (IBGE). To analyze the results, the types of POP classification were arbitrarily assigned to the following strata: (1) small (under 50,000 inhabitants), (2) medium (50,000 to 100,000 inhabitants), (3) big (100,000 to 500,000 inhabitants), and (4) metropolises (over 500,000 inhabitants).

The gross domestic product (GDP) classification was obtained from the Brazilian Institute of Geography and Statistics (https://www.ibge.gov.br, accessed on 30 May 2021), using the most recent version (2018). The cities were stratified according to their GDP in the following manner: (1) very low GDP (lower than BRL 10,000), (2) low GDP (BRL 10,001 to BRL 30,000), (3) middle GDP (BRL 30,001 to 60,000), (4) high GDP (BRL 60,001 to 100,000), and (5) very high GDP (higher than BRL 100,000).

The Municipal Human Development Index (HDI-M) of each city was extracted from the Atlas of Human Development in Brazil (http://www.atlasbrasil.org.br/, accessed on 30 May 2021) and the United Nations Development Program (https://www.br.undp.org/, accessed on 30 May 2021). According to these panels, HDI-M was stratified into (1) very low (0 to 0.499), (2) low (0.500 to 0.599), (3) medium (0.600 to 0.699), (4) high (0.700 to 0.799), and (5) very high (0.800 to 1.0). The data on epidemiological weeks were obtained from the official calendar at the following link: http://portalsinan.saude.gov.br, accessed on 30 May 2021.

Data on consolidated cases and deaths were obtained from the health information systems in the “Coronavirus Panel” of the Brazilian Ministry of Health (https://covid.saude.gov.br/). All the data contained therein are public domain information. This database is updated daily by the Ministry of Health and provides the number of confirmed cases and deaths caused by COVID-19 in Brazil, through official information sent by the State Health Departments of the 27 Brazilian Federative Units. The data provided by the states are consolidated and made public and are available every day, around 7:00 p.m. For analysis, the following variables present in the database were considered (Brazil, 2020):

1—New cases: number of new confirmed COVID-19 cases that were registered by the Municipal and State Health Departments in relation to the previous day; 

2—New deaths: number of new deaths caused by COVID-19 that were registered by the Municipal and State Health Departments in relation to the previous day. It reflects the number of deaths reported on the date on which they had laboratorial or clinical confirmation and not necessarily the date of actual occurrence of death;

3—Accumulated cases: total number of confirmed COVID-19 cases registered in the period of one week;

4—Accumulated deaths: total number of deaths caused by COVID-19 recorded in the period of one week;

5—Incidence coefficient: number of confirmed cases of COVID-19 per 100,000 inhabitants in the population residing in each geographical space in the period considered. Calculation method: (number of cases × 100,000)/(total resident population);

6—Mortality coefficient: number of deaths caused by COVID-19 per 100,000 inhabitants in the population living in each geographic space in the period considered. Calculation method: (number of deaths × 100,000)/(total resident population);

7—Lethality rate: number of confirmed deaths in relation to the total number of confirmed cases in the population residing in each geographic space in the period considered. Calculation method: (number of deaths × 100)/(number of cases).

To calculate the number of cases and deaths, as well as the incidence/mortality coefficients, and lethality rate in each epidemiological week, the sum of cases in the week was used to obtain the mean and standard deviation during every week, and then the total for the week was divided by seven days for each stratum of POP, GDP, and HDI-M.

### 2.3. Data Analysis

The number of new cases, new deaths, incidence coefficient, mortality rate, and lethality rate were observed throughout the epidemiological weeks. The period covered by this study started on the 13th week of 2020 (starting on 25 February 2020) and it ended on the 21st week of 2021 (between 23 and 29 May 2021), i.e., a 62-week period, totaling 429 days.

The four states of the region, as well as the variables POP, GDP, and HDI-M, were compared in terms of incidence coefficient, mortality rate, and lethality rate, using the areas under the curve (AUC) of the plots. The AUC values were compared by Welch’s ANOVA and the Tamhane T2 test for multiple comparisons. 

The nonlinear polynomial regression of 6th centralized order was used to observe whether the models of incidence coefficients and mortality rates adjusted to or differed from each other for POP, GDP, and HDI-M. The adjustments were compared using the Fischer test (F test) to test the null hypothesis that only one curve would fit all the conditions.

Spearman’s correlation test was used to observe the degree of correlation between the total cases and deaths in the 62 weeks with POP, GDP and HDI-M, considering each city. Spearman rho (rρ) values between 0 and 0.30 were considered as signs of negligible correlation; between 0.31 and 0.5, weak correlation; between 0.51 and 0.70, moderate; between 0.71 and 0.90, strong; and >0.90 was considered a sign of very strong correlation [9]. 

The analyses were performed using the GraphPad Prism 9.0 software, considering a significance level of 5%.

## 3. Results

The demographic profile of the Brazilian Southeast region is presented in the Appendix A. This region has 4 states with a total of 1668 cities. The average population of the city in the state of Espírito Santo showed no statistically significant differences when compared with those of Rio de Janeiro (*p* = 0.06), but the two states had larger average city populations (*p* < 0.05) than those of the other states. The HDI-M of cities in the state of São Paulo was higher (*p* < 0.0001) than the values for the other states, with the HDI-M of cities in the state of Minas Gerais being lower than the values of cities in all the other states. No statistically significant differences in HDI-M values were observed between cities in the states of Rio de Janeiro and Espírito Santo.

About 85% of cities in the Southeast region have fewer than 50,000 inhabitants, which represent 20.6% of the total state population. Only 1.26% of the cities have over 500,000 inhabitants, but they represent almost 40% of the population. The GDP is lower than BRL 10,000 in 11% of cities, and the majority (84%) showed GDP values between BRL 10,000 and BRL 60,000. All the cities with low HDI-M (4.4%) were in the state of Minas Gerais, and most of the cities (94%) had a medium or high HDI-M value.

The majority (77%) of cities in the state of Espírito Santo were small, with low GDP, but with medium to high HDI-M values. The state of Minas Gerais contains all the cities with low HDI-M (73 of 853 cities), with all of them being classified as small and with low GDP. In fact, the majority (59%) of cities in the state of Minas Gerais are small cities, with low GDP, and medium HDI-M. The state of São Paulo showed a similar profile, with 46% of the cities classified as small POP, low GDP, but with high HDI-M. In addition, the very high HDI-M cities were concentrated in the state of São Paulo and most of them reported medium population sizes and GDP values. In the state of Rio de Janeiro, great dispersion of the cities was noted, with the majority (57%) being composed of small population cities, with medium and high HDI-M, but with low GDP values.

### 3.1. Cases and Deaths in the Brazilian Southeast Region

Figure 1 shows the number of cases and deaths per epidemiological week for the Southeast region. Figure 1A shows an inflection in the curve at the 45th epidemiological week (1 to 7 November 2020), with the peak of new cases (224,189 cases) that occurred in the 67th week (4 to 10 April 2021), and the deaths (10,365 cases) in the 68th week (11 to 17 April 2021). The relationship between cases and deaths appeared to differ in the first (before November 2020) and the second (between November 2020 and May 2021) periods. The mean incidence coefficients and mortality rates (Figure 1B), considering all weeks, were 16.14 (± 9.24) and 0.56 (± 0.39), respectively. The incidence coefficient was significantly higher (Mann–Whitney test, *p* < 0.0001) in the second period (23.43 ± 6.90) than in the first period (9.73 ± 5.53). Similarly, the mortality rate was significantly higher (*p* < 0.0001) during the second period (0.78 ± 0.45) than the first period (0.37 ± 0.16).

Figure 1C revealed three distinct periods during the pandemic. The correlation between the number of new deaths and new cases was not homogeneous in these three periods. In the first period (before June 2020), the correlation was excellent (rS = 0.96, *p* < 0.0001). Very good (rS = 0.86, *p* < 0.0001) correlation was also observed in the second period (between June 2020 and March 2021), but, interestingly, only weak and non-significant correlation (rS = 0.47, *p* = 0.18) was reported in the third period (between March 2021 and May 2021). In fact, the lethality rate (Figure 1D) showed a similar pattern within the three distinct time intervals. The mean lethality rate was 6.68% (± 1.92) in the first period, decreased to 2.87% (± 0.53) in the second period (dashed lines in Figure 1D), and increased (4.55 ± 0.60%) again during the third period.

### 3.2. Cases and Deaths in the Brazilian Southeast States

The number of cases over time (Figure 2A), confirmed by the AUC value, revealed statistically significant differences (*p* < 0.0001) between the states in the Southeast region. São Paulo presented the highest AUC, followed by Minas Gerais, Rio de Janeiro, and Espírito Santo. The AUC value of deaths (Figure 2B) was also significantly higher for São Paulo (*p* < 0.0001), followed by Rio de Janeiro (*p* < 0.0001), Minas Gerais (*p* < 0.0001) and Espírito Santo (*p* < 0.0001). However, when considering the incidence coefficient (Figure 2C), the AUC value was significantly higher for Espírito Santo (*p* < 0.0001) than for the other states. There were no statistically significant differences between São Paulo and Minas Gerais (*p* = 0.5662) and both states showed higher AUC values than Rio de Janeiro (*p* < 0.0001).

Mortality, measured by the AUC value of mortality rates (Figure 2D), was higher for Rio de Janeiro (*p* < 0.05) than for the other states. Espírito Santo (*p* = 0.0360) showed higher mortality rates than São Paulo, which showed higher mortality rates than Minas Gerais (*p* = 0.0006). The segmentation into two periods (before and after November 2020) revealed that the AUC value of mortality rates was higher (*p* < 0.0001) for Rio de Janeiro, followed by Espírito Santo, São Paulo, and Minas Gerais. After the 45th week, the differences between the AUC values were reduced, and no statistically significant differences (*p* > 0.05) between the southeast states were observed. Indeed, the regression analysis showed that both the incidence coefficients and mortality rates presented different behaviors (*p* < 0.0001) across all the states.

The segmentation into two periods revealed that the evolution differed (*p* < 0.0001) among the four states in the first period (before November 2020), but the models did not differ between Minas Gerais and São Paulo (*p* = 0.87) or between Rio de Janeiro and Espírito Santo (*p* = 0.09) in the second period (after November 2020). There was evidence of strong and positive correlation between the number of cases and deaths in the states of Espírito Santo (rS = 0.77, *p* < 0.0001), Minas Gerais (rS = 0.94, *p* < 0.0001), Rio de Janeiro (rS = 0.78, *p* < 0.0001) and São Paulo (rS = 0.87, *p* < 0.0001).

### 3.3. Population Size (POP)

Figure 3 shows the number of cases and deaths, in addition to the coefficients of incidence and mortality, considering the population sizes of all the municipalities and considering all the cities of the Southern region. 

Analysis of the areas under the curve (AUC) values of the number of cases revealed that there were no statistically significant differences (*p* = 0.31) between the cities classified as “metropolis” and “big”, which showed a higher AUC value than those of the other population sizes. The lowest AUC value (*p* < 0.0001) was observed for the municipalities with medium size populations. However, the AUC values of the number of deaths showed statistically significant differences (*p* < 0.0001) among all POP sizes, being lower for the municipalities with medium size populations and higher for the metropolises. 

The AUC value of the coefficient of incidence was lower for the metropolises (*p* < 0.0001) compared to the values of the others. However, no statistically significant differences (*p* > 0.05) were observed between the other population sizes, which may indicate that the incidence was hardly affected by the size of the municipality. 

The standard mortality rate showed higher AUC values (*p* < 0.01) the higher the classification of population size was (metropolises > big > medium > small), particularly in the first period (*p* < 0.0001). However, in the second period, there were no statistically significant differences (*p* > 0.05) for mortality rate among the population sizes, indicating that in the second period, the size of the municipality did not affect the mortality rates.

Regression analysis revealed statistically significant differences (*p* < 0.0001) among all population sizes for the number of cases, deaths, and mortality rates. However, there were no statistically significant differences (*p* = 0.36) among the models for the incidence coefficient in municipalities with small and medium populations, but all the other sizes showed different models (*p* < 0.0001). There was evidence of strong positive correlation between the size of the population in the municipalities and the total number of cases (rS = 0.92, *p* < 0.0001) and deaths (rS = 0.88, *p* < 0.0001). Taken together, these results indicated that there was some influence of POP size on incidence and mortality. 

### 3.4. Gross Domestic Product (GDP)

Figure 4 shows the cases, deaths, incidence coefficients, and mortality rates ranked according to the economic profile of the municipalities, measured by the GDP. Figure 4A,B show that municipalities with middle GDP values had a higher number of cases and deaths. Analysis of the AUC values of the incidence coefficients (Figure 4C) revealed that the AUC value did not differ (*p* > 0.90) between the municipalities with high and very-high GDP, but both had higher AUC values than the other GDP strata. Those with low and middle GDP did not differ (*p* = 0.41) from the municipalities with very low GDP. 

Municipalities with very low GDP had the lowest (*p* < 0.0001) AUC values of mortality rates (Figure 4D) compared to the others. The AUC values of mortality rates of low-GDP municipalities were also lower (*p* < 0.0001) than the other strata. There were no statistically significant differences in AUC mortality values among the municipalities with middle, high and very high GDP.

The regression analysis revealed that there were no statistically significant differences (*p* < 0.0001) among the models for the incidence coefficients of the municipalities with high and very high GDP values. All the other municipalities differed from each other (*p* < 0.0001). The mortality rate models of very low and low GDP municipalities differed from each other and from the mortality rate values of all the other municipalities (*p* < 0.0001). However, there were no statistically significant differences (*p* > 0.05) among the models from middle, high and very high GDP municipalities, indicating very similar evolution of mortality patterns. In fact, there was evidence of weak positive correlation (Spearman test) between the absolute GDP values of the municipalities and the total number of cases (rS = 0.50, *p* < 0.0001) and deaths (rS = 0.49, *p* < 0.0001). 

### 3.5. Human Development Index (HDI)

Figure 5 shows the cases, deaths, incidence coefficients and mortality rates ranked according to the development of the municipalities, measured by the Municipal Human Development Index (HDI-M). Figure 5A,B show that the number of cases and deaths were influenced by the HDI-M and tended to be higher in municipalities with higher HDI-M values. In fact, AUC values of cases and deaths differed significantly (*p* < 0.0001) across all HDI-M classifications. 

Analysis of the AUC values (Figure 5C) revealed that both the incidence coefficient (*p* = 0.06) and the mortality rate (*p* = 0.99) did not differ between the municipalities with high and very high HDI-M, but both had higher AUC values than the other HDI-M classifications (*p* < 0.0001). In the second period, the municipalities with low HDI-M had a lower incidence coefficient than the other municipalities, with no statistically significant differences (*p* > 0.05) among the other HDI-M classifications. 

In this second period, the AUC mortality rate values of the municipalities with high and very high HDI-M did not differ from each other, but both showed higher AUC values than those of the other HDI-M classifications. Municipalities with medium HDI-M showed higher AUC values than those with low HDI-M (*p* < 0.0001). Regression analysis revealed that there were statistically significant differences (*p* < 0.0001) among all the models, for both the number of cases, deaths, incidence coefficients and mortality rates.

### 3.6. Lethality Rate Considering the States, POP, GDP and HDI

Figure 6 shows the lethality rate considering the states, populations, GDP, and HDI-M of the municipalities observed.

Analysis of the AUC values of the states (Figure 6A) revealed that the lethality rate in this period was significantly (*p* < 0.0001) higher in Rio de Janeiro than in the other states. São Paulo showed higher AUC values than Minas Gerais and Espírito Santo (*p* < 0.0001), and there were no statistically significant differences (*p* = 0.16) between Espírito Santo and Minas Gerais. 

The lethality rate, considering the population size (Figure 6B), showed that metropolises had higher AUC values than the other strata (*p* < 0.0001), with no statistically significant differences (*p* > 0.05) among the other population sizes. 

Considering the different GDP classifications (Figure 6C), analysis of the lethality rate AUC values showed that there were no statistically significant differences (*p* > 0.05) among the strata, except for municipalities with high GDP, which showed a higher rate than those with low GDP (*p* < 0.0001) and high GDP (*p* = 0.0019). 

There were no statistically significant differences among the municipalities with a medium, high (*p* = 0.39) and very high HDI-M value (*p* = 0.99), but all the others differed from each other. The AUC lethality rate value was higher in municipalities with a high HDI-M than those with very-high and low HDI-M classifications, with the latter being lower than the rates for all the other HDI-M classifications.

The incidence coefficient (/100.000 inhabitants) and mortality rate (/100.000 inhabitants) according to POP, HDI-M and GDP in each state is presented in Table 1. Irrespectively, the states, incidence coefficient and mortality rate showed negligible or weak correlation (rS < 0.4) with POP, GDP, and HDI-M, except for the state of Espírito Santo, where GDP showed moderate correlation (rS = 0.6, *p* < 0.0001) with the incidence coefficient. 

Lethality rates according to POP, HDI-M and GDP in each state are shown in Table 2. Only Rio de Janeiro presented weak correlation (rS = 0.4, *p* < 0.0001) between POP and lethality. In all other states, negligible correlation (rS < 0.2) was found between lethality and POP or HDI-M or GDP. 

## 4. Discussion

A large part of the Brazilian studies on the epidemiology of the pandemic was conducted in a short or very short period, often only one or two months, particularly at the beginning of the pandemic. Many of these studies presented conclusions that, in light of current knowledge, would be unreasonable, but they were important at the time when they were published [10]. In this initial period, many studies used limited temporal samples to predict or project the behavior of the pandemic, or even to observe the influence of some variables on the number of cases and deaths.

On 10 April 2020, a technical note about the disease in Brazil was released, showing that 553 deaths and 12,056 cases had been registered [11]. The highest lethality rates were observed in the Southeast region (5.5%). In the states of São Paulo and Rio de Janeiro, rates of 9.7 and 6.2 deaths per 100 thousand inhabitants, respectively, were observed, which already indicated a phase of uncontrolled acceleration of the pandemic. In fact, as shown in Figure 1D, lethality rates were high in the initial weeks, particularly between the 14th and 22nd week. 

Data of the epidemic in Brazil between February and August 2000 showed 3,817,904 cases and 120,530 deaths [12]. At that time, the Southeast and Northeast regions had the highest lethality rates. In the present study, it was possible to observe three distinct time intervals in the pandemic in the Southeast region. During the first time interval, between the 13th and 25th weeks (between March and June 2020), the pandemic was characterized by a high mortality rate, but with a low number of cases, when compared with the subsequent stages. In this period, despite the lack of tests and diagnostic certainty [6,13], it was evident that the new syndrome, with various unknown clinical aspects, could have an impact on deaths worldwide [14,15] and in Brazil [16]. 

The Southeast region could be considered a good example of the difficulty in treating the disease and its various symptoms, despite it being recognized as the region that had more health resources than many others in Brazil [17]. This fact has previously been observed when data between March 2020 and January 2021 were analyzed. All regions of the country had significant mortality rates from March 2020 onwards. Furthermore, the lack of planning before the virus arrived in Brazil was pointed out as the main cause of high mortality rates [16]. However, there are still no definitive studies about the clinical performance during these first weeks.

The second period observed in the present study, between June 2020 and March 2021, corroborates the hypothesis of the inexperience in the treatment of the disease, since the mortality rates remained stable during this long period. In this period, therefore, training and the evolution of knowledge were important for the reduction in and stabilization of death rates. Furthermore, both the decrease and variation in mortality rates in this period could, at least partly, be explained by the restriction measures. The adherence to restriction recommendations varied greatly across the country [18].

In the third period, between March and May 2021, as shown in Figure 1D, the lethality rates increased again. Figure 1C shows that there was no relationship between the number of cases and deaths in this period, as observed in previous periods. This phenomenon, with a large increase in the number of cases without any clear correlation with the increase in the lethality rate, could be explained by the appearance of more contagious variants across the country. 

The “P.1” or “Gama” strain appeared in December 2020 [16,19] or mid-November [20] in the city of Manaus-AM, and spread across Brazil, being associated with local outbreaks of great magnitude, due to its high level of transmissibility [19]. The arrival of this variant in the Southeast region may possibly explain the increase in the case lethality rate in the third period (March to May 2021). The “P.2” or “Zeta” strain was detected in the city of Rio de Janeiro in October 2020 and became the most prevalent strain in Rio de Janeiro, expanding rapidly across the entire country. Both P.1 and P.2 variants account for almost three quarters of the lineages sequenced in Brazil [21]. Therefore, both variants could help to explain the increase in the lethality rates verified in the above-mentioned period.

However, the variants P.1 and P.2 do not explain the increase in the incidence coefficients and mortality rates verified from the first week of November 2020, as shown in Figure 1B, because they were not common circulating variants at that time. However, they could have been present and active already during this period. An ancestor of P.1 was observed as far back as August 2020 and the P.1 strain in October of that year [20]. Indeed, P.1 was already present in the Southern region at the end of November 2020. 

The dynamics of the pandemic differed in the 4 states over the course of the 429 days monitored in the present study. Considering the numbers of deaths (Figure 2B), the logic of “more deaths in the largest populations” seemed to prevail, as the number of deaths in São Paulo were clearly higher than in other states. In the same figure, the inflection and sudden increase in cases close to the first week of November 2020 was also evident. When considering the proportions by population (mortality rates), the similarity between the curves was evident. However, comparison of the data obtained of the cases and deaths recorded daily proved these to be inconsistent, due to the large fluctuation in numbers.

Therefore, weekly comparison was shown to be more precise. In fact, Figure 2C,D show the differences among states more clearly. In these figures, it was possible to observe that although the state of Espírito Santo had the highest incidence coefficient, Rio de Janeiro showed the highest mortality rate.

After the first week of November 2020, the mortality rate became more uniform among the states; that is, the differences between the states occurred before November 2020. Interestingly, the analysis of the behavior of the pandemic revealed that Rio de Janeiro showed alarming lethality rates throughout the entire period studied (Figure 6A). While the states of Espírito Santo, Minas Gerais, and São Paulo maintained mortality rates in the range of 2 to 5%, in the period between June 2020 and March 2021, in Rio de Janeiro, the rates remained at significantly higher levels. Despite the alarming rates, Rio de Janeiro showed the lowest incidence coefficient among all the states in the region (Figure 2C). In Rio de Janeiro, between March and September 2020, the large number of deaths may have been caused by significant social inequality, particularly in the city of Rio de Janeiro, where over 20% of the population live in “favelas” [22]. 

Minas Gerais had the lowest mortality rate among all the states, and a lower lethality rate than São Paulo and Rio de Janeiro. Minas Gerais adopted strict measures to control the spread of the virus, particularly in the capital, where commercial establishments were closed between March and August 2020 [23]. The incidence coefficient in the capital was lower than it was in other smaller cities in this state during the period. Moreover, in Minas Gerais, the dynamics of viral propagation were noticeable not only in the capital, but also in other areas. In São Paulo, between March and July 2020, transmission of COVID-19 occurred within the capital of São Paulo and in cities surrounded by highways and then spread across the expanded metropolitan region of the city of São Paulo [24]. These authors identified the shared transport network and greater interaction between people in metropolitan areas as being the primary causes of the phenomenon. 

Although cities with over 500,000 inhabitants had a lower incidence coefficient, the mortality rate was higher in these cities in the first week of November 2020. As from the 46th week onwards, the population size did not affect the mortality coefficient, but the cities with the largest populations had the highest lethality rates (Figure 6B). The correlation between the size of Brazilian cities and the number of cases and deaths in the period between February and August 2020 showed that small cities were proportionally more affected during the initial period. However, in the long term, large cities had the highest rates of cases and deaths, a phenomenon similar to that which was observed in the present study [25]. In our study, it was not possible to decisively conclude that population size was a crucial factor in the evolution of the pandemic. 

Many studies have addressed the vulnerability of populations, as there are reports that have indicated higher incidence coefficients and mortality rates in disadvantaged communities [3,26,27]. 

GDP is the first easily understandable index that can be used to categorize the vulnerability of a population. In the Southeast region, 65% of the cities had a GDP per capita of between BRL 10,000.00 and BRL 30,000.00, and 90% of these were small cities. In total, close to 11% of these cities had a GDP lower than BRL 10,000.00 and practically all (98%) of them were extremely small. At first glance, the municipalities considered in the present study that had a GDP of between BRL 10,000.00 and BRL 60,000.00 had a much higher number of cases and deaths than the other strata of GDP (Figure 4A,B). Nevertheless, observation of the incidence coefficients (Figure 4C) and mortality rates (Figure 4D) showed another reality. 

Interestingly, the “poorest” municipalities (GDP lower than BRL 10,000.00) had lower incidence coefficients and mortality rates. Municipalities with higher GDP values, in general, had higher incidence coefficients than municipalities with lower GDP values, but no evidence of this effect was shown in the mortality rates. In fact, in the state of Rio de Janeiro, the lethality rate was hardly affected by the GDP value and there was evidence of weak correlation between GDP and the total numbers of cases and deaths. These results were not compatible with those of various other studies observed in the literature. The poorest population would be more susceptible during health crises, as there would be more unemployment, weakening of social safety nets and less access to health services [26], in addition to a larger proportion of people living in dwellings with excessively high density [13]. 

Lower per capita income and higher levels of poverty and illiteracy were associated with more cases and deaths in municipalities that had over 100,000 inhabitants [28]. On the contrary, other authors observed that municipalities with higher incomes had higher incidence coefficients and mortality rates [29]. In a similar manner to the data presented here, other authors did not observe an important correlation between GDP and the infection rate [24]. 

These results must obviously be interpreted with caution, as they were relative to an assessment of municipalities and not individuals. Furthermore, the dimensions of well-being and vulnerability are not evaluated by the GDP alone. This explains the importance of understanding the HDI-M in the evolution of the pandemic. This index measures social, cultural, and political characteristics that are capable of influencing the quality of life of the population.

The influence of the so-called social determinants on the evolution of the pandemic has been recognized in many studies [3,13,28,30]. Social determinants, such as HDI, distance to the hospital (rural and less developed areas), education, type of hospital and ethnicity, may be more important than comorbidities, and as important as biological factors, in understanding the outcome of the disease [31]. However, by using the HDI-M as a social determinant, the present study showed that a trend towards higher incidence coefficients (Figure 5C) and mortality rates (Figure 5D) was evident in cities that had a high or extremely high HDI-M. The relations between cases, deaths and the HDI-M values was shown to be positive, and the lethality rate was also shown to have a direct relationship with this parameter. These results contradicted the idea that the best state of human development in the municipality contributed decisively to the lower impact of the pandemic on the city.

Data between February and June 2020 showed that, despite the declaration by politicians and some media outlets that “the COVID-19 virus does not discriminate”, the disease is not “socially neutral”; disadvantaged communities have been disproportionately affected [27]. In a similar manner, other authors, by analyzing data between February and September 2020, reported that the disease was considered to be “equalizing” in the beginning [30]. Its evolution was, however, significantly impacted by social inequalities, with its prevalence and severity being linked to poverty, lower educational levels, less access to health services, economic insecurity and precarious neighborhood and housing conditions.

Data collected up to October 2020 showed that socioeconomic aspects were poor predictors of the disease; however, lower HDI values indicated worse evolution in municipalities that had over 100,000 inhabitants [28]. However, for the Southeast region, particularly for the states of São Paulo and Rio de Janeiro, the HDI was not considered as decisive. In a similar manner, other authors have observed that higher levels of social inequality measured by the GINI index may be directly related to the lethality rate in the municipalities of the North and Northeast regions [26]. These authors failed to relate this same index to deaths and cases in the Southeast region. Therefore, the direct causal relationship between socioeconomic aspects and the progression of the pandemic continues to be controversial and has shown to be dependent on factors such as the period evaluated, the territorial region considered and other biological aspects (sex, age, ethnicity, etc.). 

Some inherent biases in the model used in the present study must be noted. The data were obtained directly from the database of the Department of Informatics of the SUS (DATASUS), which is not a foolproof system [5]. Consequently, it is not possible to guarantee the registration of all cases of and deaths caused by COVID-19. 

There has been a lack of tests and follow-up reports of possible cases [6,13] since the beginning of the pandemic, resulting in underreporting. Furthermore, health records are updated and entered into electronic systems after the events [12], particularly after weekends. 

Moreover, the system often fails to capture all the details, due to the incompleteness and inaccuracy of the hospital records. Even simple data such as comorbidity exams, laboratory tests, ethno-racial classification, sex, age, and ICU admission are absent in some cases [29]. Therefore, the logical option was to use the data in the simplest and most reliable manner possible, considering only registered cases and deaths, without considering variables such as sex, age and comorbidities. The addition of these variables could certainly enrich the data and help explain part of the results obtained in the present study.

Another example of bias, also observed by other scholars [3], was the reliability of information obtained from sociodemographic data. The municipalities may have undergone sociodemographic changes during the period, and these would influence the results for the locality. However, we assumed that there were no significant changes in this profile over the course of ten years, as did Ref. [3]. Unfortunately, it is not possible to definitively assess the impact of all the factors on the results obtained, as there are insufficient data in the system.

The present study showed that the parameters observed in isolation showed results that were—at least partly—contradictory to those observed in the literature. Multivariate analysis that considers all the population, economic (GDP) and development (HDI) parameters would obviously be more adequate. However, due to the different characteristics of the municipalities in the region, it was not possible to perform this type of stratification, as shown in Table 1 and Table 2. 

Therefore, further studies on the real impact of socioeconomic and demographic determinants should be conducted to increase the body of knowledge about the evolution of the pandemic, with the aim of improving preparation for possible future events.

## 5. Conclusions

It was possible to conclude that there were three different time intervals of the pandemic in the Southeast region throughout the period studied. The incidence coefficients and mortality rates were affected by time and showed different patterns in relation to the states. POP, GDP, and HDI-M had little influence on the above-mentioned parameters. Furthermore, lethality reached alarming levels in the first 25 weeks of the pandemic, but these levels stabilized over the course of time.

## Figures and Tables

**Figure 1 ijerph-19-14459-f001:**
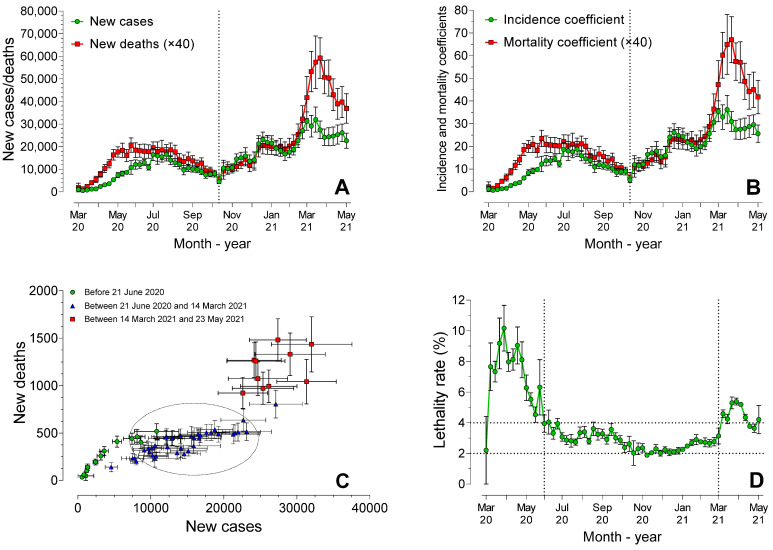
Weekly mean (± standard error) of the number of new cases and new deaths (**A**), coefficient of incidence and coefficient of mortality (**B**), correlation between number of new cases and new deaths (**C**) and lethality rate (**D**), throughout this period.

**Figure 2 ijerph-19-14459-f002:**
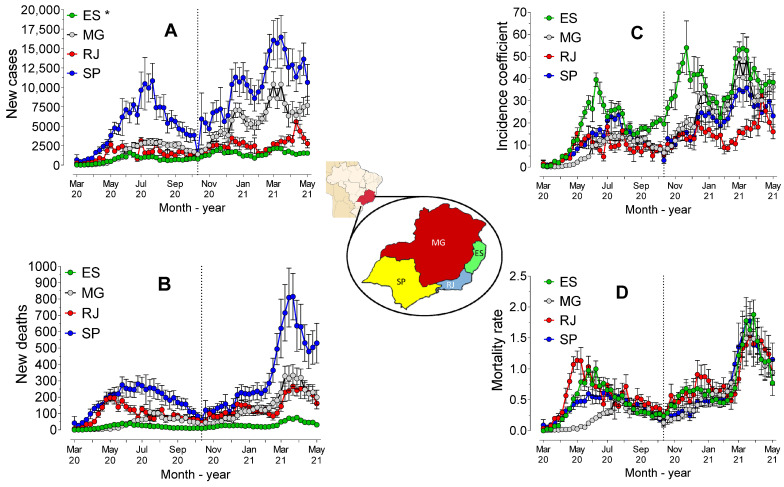
Mean (± standard error) of cases (**A**) and new deaths (**B**), incidence coefficient (**C**) and mortality rate (**D**), over the course of 62 weeks of the pandemic in the 4 states of the Southeast region. * ES = Espírito Santo, MG = Minas Gerais, SP = São Paulo; RJ = Rio de Janeiro.

**Figure 3 ijerph-19-14459-f003:**
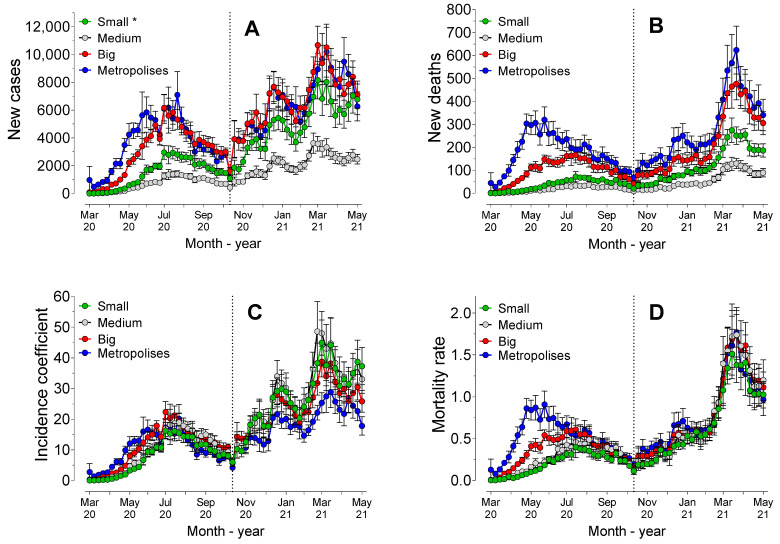
Mean (±standard error) of cases (**A**) and new deaths (**B**), incidence coefficients (**C**), and mortality rates (**D**) during the 62 weeks of the pandemic, considering the four strata of population sizes of the municipalities. * Small cities (under 50,000 inhabitants); medium cities (50,000 to 100,000 inhabitants); big cities (100,000 to 500,000 inhabitants); and metropolises (over 500,000 inhabitants).

**Figure 4 ijerph-19-14459-f004:**
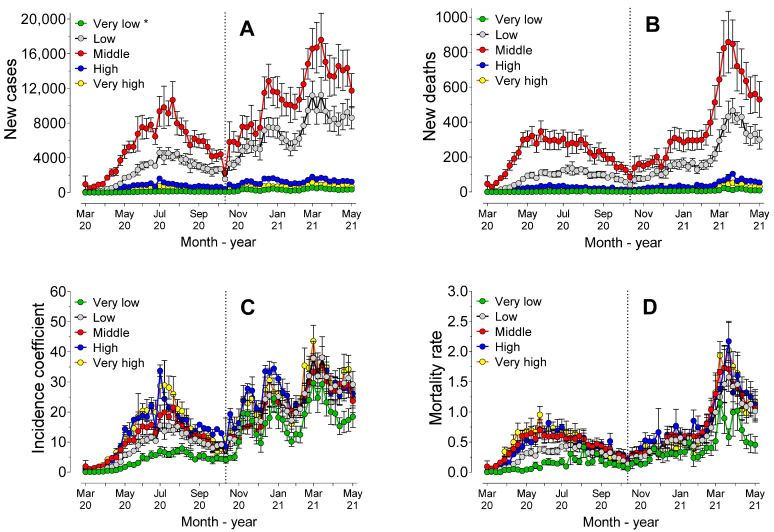
Mean (standard error) of cases (**A**) and new deaths (**B**), incidence coefficient (**C**) and mortality rate (**D**), over the course of 62 weeks of the pandemic, considering the gross domestic product (GDP) strata of the cities. * Very low GDP (lower than BRL 10,000); low GDP (BRL 10,001 to BRL 30,000); middle GDP (BRL 30,001 to 60,000); high GDP (BRL 60,001 to 100,000); and very high GDP (higher than BRL 100,000).

**Figure 5 ijerph-19-14459-f005:**
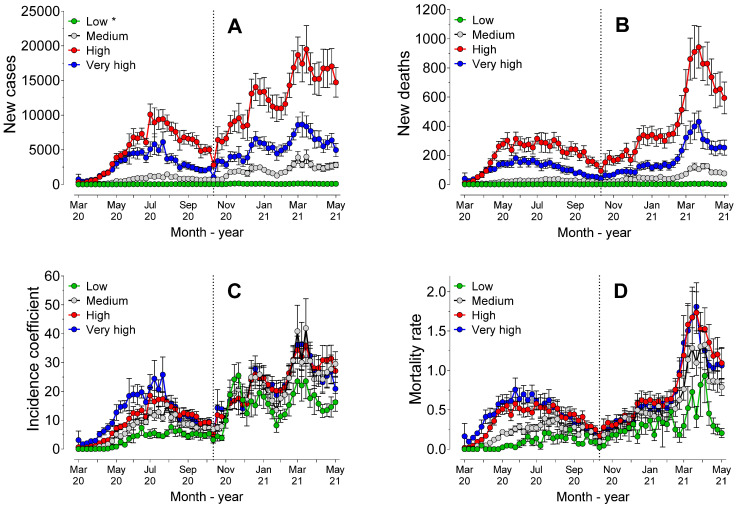
Mean (95%CI) of cases (**A**) and new deaths (**B**), incidence coefficient (**C**) and mortality rate (**D**), over the course of 62 weeks of the pandemic, according to the Municipal Human Development Index (HDI-M) of the cities. * Very low HDI-M (0 to 0.499); low HDI-M (0.500 to 0.599); medium HDI-M (0.600 to 0.699); high HDI-M (0.700 to 0.799); and very high HDI-M (0.800 to 1.0).

**Figure 6 ijerph-19-14459-f006:**
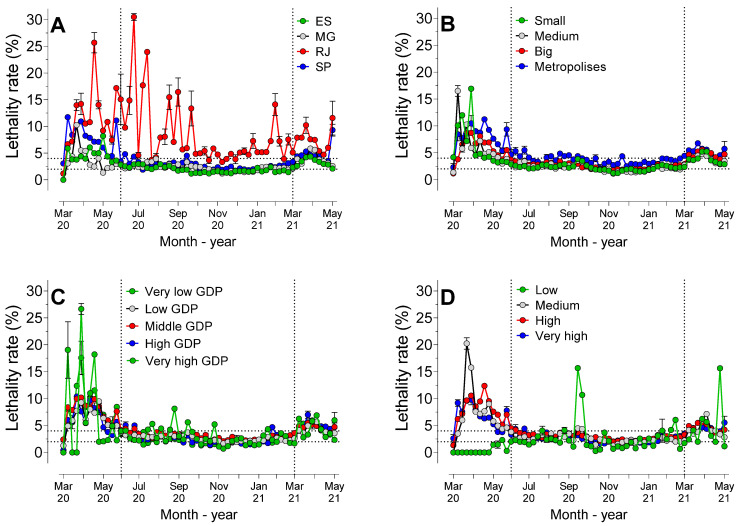
Mean lethality rate (95%CI) considering the following: (**A**) states; (**B**) population size; (**C**) gross domestic product (GDP); and (**D**) Municipal Human Development Index (HDI-M), over the course of 62 weeks of the pandemic. Legend (**A**)—ES = Espírito Santo; MG = Minas Gerais; SP = São Paulo; and RJ = Rio de Janeiro. Legend (**B**)—small cities (under 50,000 inhabitants); medium cities (50,000 to 100,000 inhabitants); big cities (100,000 to 500,000 inhabitants); and metropolises (over 500,000 inhabitants). Legend (**C**)—very low GDP (lower than BRL 10,000); low GDP (BRL 10,001 to BRL 30,000); middle GDP (BRL 30,001 to 60,000); high GDP (BRL 60,001 to 100,000); and very high GDP (higher than BRL 100,000). Legend (**D**)—very low HDI-M (0 to 0.499); low HDI-M (0.500 to 0.599); medium HDI-M (0.600 to 0.699); high HDI-M (0.700 to 0.799); and very high HDI-M (0.800 to 1.0).

**Table 1 ijerph-19-14459-t001:** Incidence coefficient per 100,000 inhabitants (mortality rate per 100,000 inhabitants), over the 62 weeks in the 4 states of the Southeast region, according to population size (POP); gross domestic product (GDP); and Municipal Human Development Index (HDI-M), also over the course of 62 weeks of the pandemic.

		Very Low GDP			Low GDP				Middle GDP			High GDP			Very High GDP		
State	POP	Low HDI	Medium HDI	High HDI	Low HDI	Medium HDI	High HDI	Very High HDI	Medium HDI	High HDI	Very High HDI	Medium HDI	High HDI	Very High HDI	Medium HDI	High HDI	Very High HDI
ES	Small	--	--	--	--	10,316.2 (236.5)	12,896 (232)	--	12,173.8 (126.1)	13,681.5 (313.9)	--	16,106.9 (441.6)	--	--	13,091.8 (383.3)	--	--
	Medium	--	--	--	--	--	9960.1 (225.5)	--	9741.9 (297.8)	--	--	--	--	--	--	--	--
	Big	--	--	--	--	--	10,930.7 (280)	11,992.6 (300.7)	--	14,006.7 (204.5)	--	--	--	14,291.8 (325.9)	--	--	--
	Metrop.	--	--	--	--	--	--	--	--	11,905.5 (259.7)	--	--	--	--	--	--	--
MG	Small	3902.5 (87.6)	4560.9 (117)	4195.7 (79)	4878.5 (83.9)	6364.9 (159.1)	7229.2 (189.1)	--	7777.5 (195.4)	9396.5 (237)	--	7783 (187.8)	7495.8 (163)	--	7324.5 (218.7)	14,606.2 (206.2)	--
	Medium	--	3102.7 (164.4)	2020.5 (58.6)	--	5111.1 (134.1)	8439.6 (216.4)	--	--	9417.3 (224.6)	--	--	12,754.1 (160.9)	--	--	--	16,162 (203.4)
	Big	--	--	--	--	4499.2 (152)	7133.6 (203.9)	--	--	8892.7 (242.8)	--	--	--	--	--	--	--
	Metrop.	--	--	--	--	--	5828 (284.2)	--	--	9508 (283.9)	8147.2 (201.8)	--	--	--	--	--	--
RJ	Small	--	--	--	--	6140.2 (215.6)	9383 (264.6)	7539.5 (296.8)	12,461.4 (336.4)	6793.7 (175)	--	5423.4 (155.2)	9724.5 (225.8)	--	14,309.5 (72)	7356.8 (233.6)	--
	Medium	--	--	--	--	7655.7 (274.3)	6260.4 (229.9)	--	--	5049.9 (280.5)	--	--	--	--	--	--	--
	Big	--	--	--	--	2927.1 (192.9)	5452.2 (259.1)	--	--	6441.2 (304.6)	--	--	7861.9 (262.4)	--	--	5528.9 (272.3)	--
	Metrop.	--	--	--	--	3907.6 (127.8)	3263.1 (204.3)	--	--	4509.3 (362.5)	--	--	4776.7 (246.9)	7273.6 (366.8)	--	--	--
SP	Small	--	5947 (115.2)	8693 (241.1)	--	6669.9 (165.2)	7229.5 (201.2)	7629.5 (299.8)	8720.1 (219.9)	8184.1 (217.7)	7910.8 (231.8)	7519.6 (252.3)	8308.6 (220.8)	--	--	10,762.6 (181)	--
	Medium	--	--	--	--	--	7036.8 (221)	--	--	8134.4 (215.9)	9653.3 (265.7)	--	7180 (238.2)	--	--	6607.3 (186.7)	9043.8 (170.2)
	Big	--	--	5311.5 (212.1)	--	--	5991.9 (227.2)	--	4968.9 (215.6)	7179.7 (251.9)	8968.3 (235.5)	--	9869.8 (216.4)	9139.2 (348.3)	--	9582.7 (311.9)	9312.3 (281.9)
	Metrop.	--	--		--	--	--	--	--	5220 (283.3)	6722.2 (256.8)	--	--	7400.2 (313.8)	--	5258.6 (293.2)	--

Legend: states: ES = Espírito Santo; MG = Minas Gerais; SP = São Paulo; and RJ = Rio de Janeiro. POP classification: small cities (under 50,000 inhabitants); medium cities (50,000 to 100,000 inhabitants); big cities (100,000 to 500,000 inhabitants); and metropolises (over 500,000 inhabitants). GDP classification: Very low GDP (lower than BRL 10,000); low GDP (BRL 10,001 to BRL 30,000); middle GDP (BRL 30,001 to 60,000); high GDP (BRL 60,001 to 100,000); and very high GDP (higher than BRL 100,000). HDI-M classification: very low HDI-M (0 to 0.499); low HDI-M (0.500 to 0.599); medium HDI-M (0.600 to 0.699); high HDI-M (0.700 to 0.799); and very high HDI-M (0.800 to 1.0).

**Table 2 ijerph-19-14459-t002:** Lethality rate over the 62 weeks in the 4 states of the Southeast region, according to population size (POP); gross domestic product (GDP); and Municipal Human Development Index (HDI-M), also over the course of 62 weeks of the pandemic.

		Very Low GDP			Low GDP				Middle GDP			High GDP			Very High GDP		
State	POP	Low HDI	Medium HDI	High HDI	Low HDI	Medium HDI	High HDI	Very High HDI	Medium HDI	High HDI	Very High HDI	Medium HDI	High HDI	Very High HDI	Medium HDI	High HDI	Very High HDI
ES	Small	--	--	--	--	0.24	0.23	--	0.13	0.31	--	0.44	--	--	0.38	--	--
	Medium	--	--	--	--	--	0.23	--	0.30	--	--	--	--	--	--	--	--
	Big	--	--	--	--	--	0.28	0.30	--	0.20	--	--	--	0.33	--	--	--
	Metrop.	--	--	--	--	--	--	--	--	0.26	--	--	--	--	--	--	--
MG	Small	0.09	0.12	0.08	0.08	0.16	0.19	--	0.20	0.24	--	0.19	0.16	--	0.22	0.21	--
	Medium	--	0.16	0.06	--	0.13	0.22	--	--	0.22	--	--	0.16	--	--	--	0.20
	Big	--	--	--	--	0.15	0.20	--	--	0.24	--	--	--	--	--	--	--
	Metrop.	--	--	--	--	--	0.28	--	--	0.28	0.20	--	--	--	--	--	--
RJ	Small	--	--	--	--	0.22	0.26	0.30	0.34	0.18	--	0.16	0.23	--	0.07	0.23	--
	Medium	--	--	--	--	0.27	0.23	--	--	0.28	--	--	--	--	--	--	--
	Big	--	--	--	--	0.19	0.26	--	--	0.30	--	--	0.26	--	--	0.27	--
	Metrop.	--	--	--	--	0.13	0.20	--	--	0.36	--	--	0.25	0.37	--	--	--
SP	Small	--	0.12	0.24	--	0.17	0.20	0.30	0.22	0.22	0.23	0.25	0.22	--	--	0.18	--
	Medium	--	--	--	--	--	0.22	--	--	0.22	0.27	--	0.24	--	--	0.19	0.17
	Big	--	--	0.21	--	--	0.23	--	0.22	0.25	0.24	--	0.22	0.35	--	0.31	0.28
	Metrop.	--	--	--	--	--	--	--	--	0.28	0.26	--	--	0.31	--	0.29	--

Legend: states: ES = Espírito Santo; MG = Minas Gerais; SP = São Paulo; and RJ = Rio de Janeiro. POP classification: small cities (under 50,000 inhabitants); medium cities (50,000 to 100,000 inhabitants); big cities (100,000 to 500,000 inhabitants); and metropolises (over 500,000 inhabitants). GDP classification: very low GDP (lower than BRL 10,000); low GDP (BRL 10,001 to BRL 30,000); middle GDP (BRL 30,001 to 60,000); high GDP (BRL 60,001 to 100,000); and very high GDP (higher than BRL 100,000). HDI-M classification: very low HDI-M (0 to 0.499); low HDI-M (0.500 to 0.599); medium HDI-M (0.600 to 0.699); high HDI-M (0.700 to 0.799); and very high HDI-M (0.800 to 1.0).

## Data Availability

Publicly available datasets were analyzed in this study. This data can be found here: https://covid.saude.gov.br; https://www.ibge.gov.br; http://www.atlasbrasil.org.br; https://www.br.undp.org; http://portalsinan.saude.gov.br.

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
