# Peer review of "Influence of Population Size, the Human Development Index and the Gross Domestic Product on Mortality by COVID-19 in the Southeast Region of Brazil"

_ijerph, 2022, doi:10.3390/ijerph192114459_

Round 1

Reviewer 1 Report

An interesting paper with a unique perspective on the COVID-19 situation in Brazil. However, since there are many enumerations of numbers, it is difficult for readers to grasp and understand the analysis results. For example, it is better to increase the number of graphs and tables to make it easier to grasp visually. The current style of writing frustrates readers before they even read the details of this paper. If you want me to revise this paper on this point, I'd like to read it again. I'm looking forward to it after the fix.

Author Response

We would like to thank the Reviewer´s suggestions and criticism. We agree that there are too many numbers, and it is difficult to understand all the results, at some point. We have kept the Figures at a minimum necessary to understand the results, avoiding repetition, and to keep the text clarity. We could not include more graphics to represent the results, but we will be glad to split the figures, since they represent four different characteristics. If the Editor judges necessary to change the graphics format or to include more tables, we will be glad to change the manuscript. We have revised all the manuscript text to improve its re

Regarding the “English changes required”, we would like to inform that the manuscript was extensively revised before submission, as certified by Mrs. Margery J. Galbraith (attached file). We have promoted minor changes in the manuscript text, and we hope you find them appropriated. Otherwise, we will be glad to change it even more.

ading. We hope you find it acceptable.

Reviewer 2 Report

The authors have done tremendous work, however, overall I do not feel the manuscript, as is, is not suitable for publication. It is not clear in the introduction why the Southeast region was selected. What are the importance and rationale of the study? The aim is confusing - the authors could consider aligning it with the study methods and results.

Some concepts (e.g., new cases, incidence, mortality rates) were not clearly defined or not described at all. 

In the results, it is not clear what is the rationale for comparing cities in terms of population sizes. It is likely that differences will be there statistically significantly speaking. I was also not sure what was the purpose of testing the correlation between new cases and new deaths? - while this was not the aim of the study. 

Some minor comments:

Please spell out all abbreviations when first time mentioned

Not clear why POP was categorized in such a way. Similar to the GDP, please explain the rationale for categorizing it this way.

Definitions for new cases and new deaths are slightly confusing. Some clarification could improve the readability of the manuscript.

Instead of "lethality rate” the authors can use case-mortality rate, as it is a commonly accepted term.

It is not clear how weekly rates were calculated: did the authors average the estimates? Or summed them?

This is slightly confusing “The 62-week period studied covers the time elapsed between the 13th week of 2020 (that started on February 25, 113 2020) until the 21st week of 2021 (between 23 and 29 May 2021)” the numbering of weeks makes little sense

Author Response

The authors have done tremendous work, however, overall I do not feel the manuscript, as is, is not suitable for publication.

We would like to thank the Reviewer´s suggestions and criticism. Indeed, it was a tremendous work and a lot of numbers to present.

It is not clear in the introduction why the Southeast region was selected.

We have included more information in the introduction regarding the Southeast Brazilian region, which is the most populous and the most important regarding the country economy.

What are the importance and rationale of the study? The aim is confusing - the authors could consider aligning it with the study methods and results.

We have revised and changed the study’s aim in order to align it with the study methods and results, as recommended.

Some concepts (e.g., new cases, incidence, mortality rates) were not clearly defined or not described at all. 

We tried to keep the manuscript text short and some concepts were in fact insufficiently explained. We revised all the text and improved the definitions. 

In the results, it is not clear what is the rationale for comparing cities in terms of population sizes. It is likely that differences will be there statistically significantly speaking. I was also not sure what was the purpose of testing the correlation between new cases and new deaths? - while this was not the aim of the study. 

Regarding the comparisons among cities with different population sizes, we used it to show that, despite the obvious trend, i.e., the more people the more deaths, considering the incidence coefficient and mortality rate the obvious trend disappeared. Thus, in spite of the city population size (big or small) the proportion were very similar, especially in the second period. Curiously, the statistical differences were not there.

We consider the Figure 1C (that shows the correlation between new cases and new deaths) as one of the most interesting in the study. It shows three different periods during the pandemic. In the first one (before 26th week), the more the cases the more the deaths, probably due to the inability to treat the patients and to control the disease spread. During the second period (26th to 64th weeks), despite the increasing number of cases, deaths were not directly proportional. It could be a reflex of an improved medical care. The last period (after 65th week) showed a big number of cases with a direct proportional number of deaths. During this period, new strains emerged, affecting the health system. Thus, the correlation between new cases and new deaths addressed an important issue.

Some minor comments:

Please spell out all abbreviations when first time mentioned

We have checked all the abbreviations according to the Referee´s recommendation.

Not clear why POP was categorized in such a way. Similar to the GDP, please explain the rationale for categorizing it this way.

At first, we used both categories of population size and GDP as recommended by the Brazilian Institute of Geography and Statistics. However, it uses too many classifications in the strata, making very difficult to analyze and represent the results. Thus, we have reduced the categories, in a similar fashion with other authors.

Definitions for new cases and new deaths are slightly confusing. Some clarification could improve the readability of the manuscript.

We have used the definitions provided by the "Coronavirus Panel" of the Brazilian Ministry of Health. However, we have changed it to improve clarity.

Instead of "lethality rate” the authors can use case-mortality rate, as it is a commonly accepted term.

The "lethality rate” was a definition provided by the "Coronavirus Panel" of the Brazilian Ministry of Health. The term case-mortality rate is very unusual in the indexed literature (Medline). Thus, we have decided to keep the term. However, if the Editor judges necessary to change, we will be glad to do so.

It is not clear how weekly rates were calculated: did the authors average the estimates? Or summed them?

We have included a better description on this issue. The mean and standard deviation of the number of cases and deaths, as well as the incidence coefficients/mortality rates, and lethality rate, was obtained dividing the total (of cases, deaths, etc.) by seven (days).

This is slightly confusing “The 62-week period studied covers the time elapsed between the 13th week of 2020 (that started on February 25, 113 2020) until the 21st week of 2021 (between 23 and 29 May 2021)” the numbering of weeks makes little sense

We have changed this paragraph to clarify it. 

Round 2

Reviewer 1 Report

As I commented in the previous review, it is difficult for readers to understand the results even if the results are reported only in sentences and figures. I think it's easier to understand if you use a table instead of a graph.

In particular, "4. Discussion" has a very long chapter, and I cannot see the point.

Author Response

As I commented in the previous review, it is difficult for readers to understand the results even if the results are reported only in sentences and figures. I think it's easier to understand if you use a table instead of a graph.

We would like to thank the Referee for the considerations and the time/efforts to help us to improve the manuscript text. We strong believe that the IJERPH readers will have the skills necessary to understand the figures. We did put enormous efforts to present the data in tables, as suggested. However, they are also very big and probably more difficult to observe the trends for each variable. If every single figure were represented in tables, we should have six tables with 63 lines and at least 16 (or more!) columns. The most import contribution of each figure is to show the trends along time and not the individualized results.

If the Editors judges that is necessary to present the data in tables, we have included some of them in the supplementary material. If not necessary, we will remove them from the supplementary section. We decide to keep the figures as previously stated.

In particular, "4. Discussion" has a very long chapter, and I cannot see the point.

In the Discussion section we have explored all the important aspects that influencing the study. We agree that is a long text. Thus, we have removed some more generic statements, not directly linked with the results, to shorten the chapter. We hope you find acceptable.